# Study on Mechanisms of NO$_x$ Formation and Inhibition during the Combustion of NH$_3$/CH$_4$ and NH$_3$/CO Mixtures

Yongbo Du [1], Siyu Zong [1], Chang'an Wang [1], Yongguan Wang [1], Qiang Lyu [1,*], Yaodong Da [2,*] and Defu Che [1]

[1] School of Energy and Power Engineering, Xi'an Jiaotong University, Xi'an 710049, China; ybdu_boilerynwa@xjtu.edu.cn (Y.D.); fate314@stu.xjtu.edu.cn (S.Z.); changanwang@mail.xjtu.edu.cn (C.W.); wyg522881339@stu.xjtu.edu.cn (Y.W.); dfche@mail.xjtu.edu.cn (D.C.)

[2] China Special Equipment Inspection and Research Institute, Beijing 100029, China

\* Correspondence: qianglv@xjtu.edu.cn (Q.L.); dayaodong@csei.org.cn (Y.D.)

**Abstract:** Ammonia is an ideal renewable, carbon-free fuel and hydrogen carrier, which produces nitrogen and water after complete combustion in the presence of oxygen. However, ammonia has low reactivity, slow flame-propagation speed, and carries risks of high nitrogen oxide (NO$_x$) emissions. Co-firing ammonia with an industrial by-product gas (with CH$_4$ and CO being the main combustible materials) is a cost-effective and convenient method of improving the combustion characteristics of ammonia, but attention still needs to be paid to the NO$_x$ generation. Currently, the research on NO$_x$ formation during co-firing of ammonia with other fuel gases is still insufficient. In this study, a high-temperature furnace reaction system was used to investigate the NO$_x$ formation and inhibition mechanisms during the combustion of NH$_3$/CH$_4$ and NH$_3$/CO mixtures. By varying the ammonia blending ratio, excess air coefficient ($\alpha$), temperature, residence time, and fuel concentration, the key factors influencing NO$_x$ generation and inhibition were further analyzed. The results showed that when $\alpha$ was no less than 1, the production of NO$_x$ initially increased and then decreased with an increasing proportion of ammonia in the fuel gas. Within the temperature range of 900 °C to 1500 °C, the amount of NO$_x$ generated during the combustion of the mixed gas gradually decreased with the increase in temperature. Under the conditions of NH$_3$/CH$_4$ and NH$_3$/CO, the emissions of NO$_x$ were higher than those during pure ammonia combustion.

**Keywords:** ammonia; nitrogen oxide; blending combustion; formation mechanism

## 1. Introduction

With the rapid development of modern society, the energy crisis and environmental issues are becoming increasingly serious. It is therefore urgent to reduce the use of fossil fuels as well as to develop renewable energy and clean energy. China has put forward the 'carbon peaking and carbon neutrality' goals to limit CO$_2$ emissions. The disadvantages of renewable energy such as photovoltaic and wind power include difficulties with storage and low efficiency [1,2]. Hence, some scholars have used unstable photoelectric and wind power to produce hydrogen by water electrolysis for storage [3,4]. However, hydrogen is explosive, and also expensive to store and transport [5,6]. Ammonia is an ideal renewable carbon-free fuel and hydrogen carrier, whose combustion with oxygen produces only nitrogen and water, without emitting SO$_x$ or particulate matter [7,8]. If those green hydrogens and the nitrogen separated from air are used to synthesize ammonia, complete carbon-free emissions can be achieved [9–11]. In comparison to hydrogen, ammonia is easy to liquefy, and the cost of storage and transportation is low [12]. Ammonia has a pungent smell, so it is easy to detect leaks and avoid danger [13]. Liquid ammonia has high volume energy density [14], and ammonia is safe because it is not easy to ignite at ambient temperature [13].

However, compared with conventional hydrocarbon fuels, pure ammonia has a slower laminar combustion rate and is harder to ignite in air [15,16]. In practice, it is challenging

to achieve complete combustion of ammonia, and the combustion process has a risk of exceeding $NO_x$ emissions [17]. Blending known fuels with other fuels for combustion is a common practice [18,19]. Co-firing ammonia with another combustible gas is a potential approach to improving the combustion rate and performance of ammonia, which had been studied by some scholars. Ryu et al. [20] found that the mixed combustion of ammonia and hydrogen produced by the decomposition of ammonia by catalysts can improve the combustion performance of ammonia engines and reduce exhaust emissions. Ichikawa et al. [21] studied the laminar flame propagation characteristics of an ammonia–hydrogen mixture and found that when the volume fraction of $H_2$ mixing reached 40%, the laminar flame velocity of the mixture reached 35 cm/s, which was roughly equivalent to the laminar flame velocity of $CH_4$ when it was burned in air. Li et al. [6] also found a similar variation trend in laminar flame velocity. Yasiry et al. [22] also found that the laminar combustion velocity increased with the increase in hydrogen fraction.

In order to reduce costs, ammonia can be mixed with some by-product gases produced in agricultural or industry, such as landfill gas, steel gas, or biogas. The main flammable component of these gases is $CH_4$ or CO. One study found that when ammonia was mixed with conventional hydrocarbon fuels, the flame speed, heat release rate, and radiation intensity of the fuel were higher than that of pure ammonia [3]. Okafor et al. [23] conducted experimental and numerical studies on methane–$NH_3$–air premixed flames, and found that the unstretched laminar combustion velocity of the mixture decreased with the increase in ammonia concentration. Tian et al. [24] conducted premix combustion experiments and modeling studies on $NH_3/CH_4/O_2/Ar$ under 11 kinds of excess air coefficients, and then proposed a comprehensive chemical mechanism of combustion of fuel mixtures with different ratios of ammonia/methane. Xiao et al. [25] studied the ignition process of 60% $NH_3$/40% $CH_4$ fuel mixture by using a shock tube, and found that the ignition performance of $NH_3/CH_4$ was significantly improved with the increase in temperature, pressure, and methane mixing ratio, but the influence of the excess air coefficient on ignition performance was not obvious. Wang et al. [26] conducted premixed combustion experiments on $NH_3$/syngas/air, $NH_3$/CO/air, and $NH_3/H_2$/air, and measured their laminar combustion velocities. They found that with the addition of $H_2$ and CO, the decomposition rate of ammonia was significantly increased. Han et al. [14] measured the laminar flame velocity of $NH_3$/CO at different mixing ratios and excess air coefficients in a heat-flow furnace. It was found that with the increase in $NH_3$, the laminar flame velocity of the $NH_3$ and CO mixture first increased and then decreased. When the volume fraction of $NH_3$ was 85%, the laminar flame velocity of mixture was the highest, and was much higher than that of either pure $NH_3$ or pure CO.

In comparison to combustion efficiency, research on NO formation during co-firing $CH_4$ and CO with $NH_3$ is insufficient. Zieba et al. [27] and Wang et al. [26] found that the $NO_x$ emission increased after adding doped combustion gas to ammonia. Okafor et al. [28] realized the ideal combustion characteristics of $NH_3/CH_4$ in gas turbines by using rich–poor staged combustion, with low $NO_x$ and CO emissions. Zhang et al. [29] experimentally studied the NO emission characteristics of $NH_3/CH_4$/air on premixed cyclone burners. The results showed that the $NO_x$ emission was the highest when the calorific value ratio of $NH_3$ in the mixed fuel was 0.5. Li et al. [30] and Honzawa [31] paid attention to the influence of free-radical concentration on $NO_x$ emission. However, the formation and inhibition mechanisms of $NO_x$ are still unclear.

In this paper, the NO generation characteristics of $NH_3$ mixed with CO or $CH_4$ was studied. With adopting a high-temperature furnace reaction system, experiments were conducted on the combustion of $NH_3/CH_4/Ar$ mixture and $NH_3/CO/Ar$ mixture under various operating conditions. The effects of ammonia ratio, excess air coefficient ($\alpha$), temperature, residence time, and fuel concentration on NO formation were investigated and are discussed in detail. This will help guide the clean and efficient utilization of ammonia, which is a new renewable energy source, and can turn industrial by-products into wealth.

## 2. Experimental System and Working-Conditions

### 2.1. Experimental System and Method

The schematic diagram and physical objects of the experimental system are shown in Figure 1. Argon, ammonia, oxygen, methane, carbon monoxide, and other gases used in the experiment were stored in high-pressure gas cylinders, and the pressure-reducing valve was used to control the opening of the gas cylinder. Air flow was controlled by the flow meter and D08-3E flow indicator with an accuracy of ±0.5%.

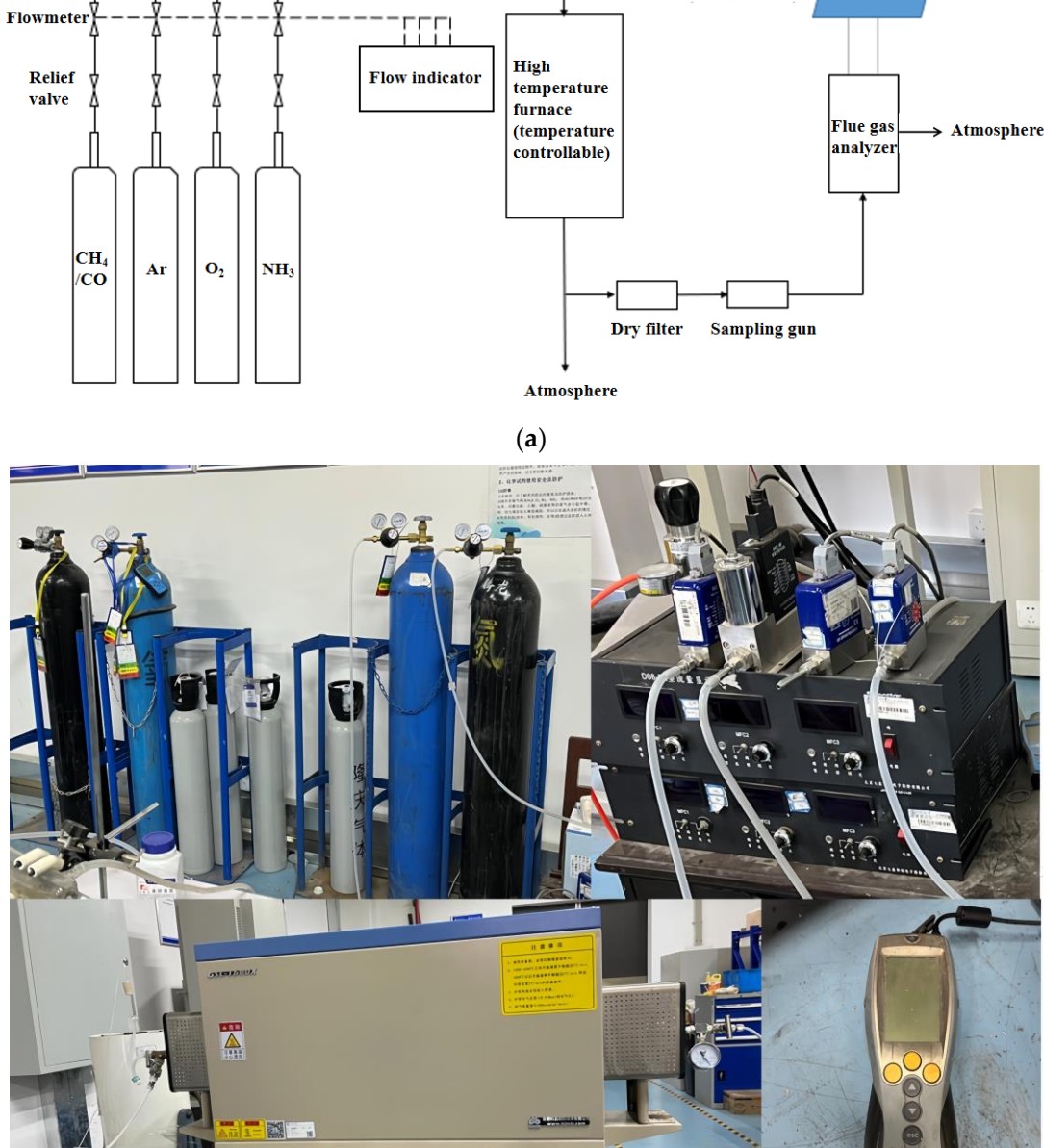

(**a**)

(**b**)

**Figure 1.** High-temperature gas combustion system. (**a**) Schematic diagram and; (**b**) Physical image objects.

The gas was mixed and then supplied to a corundum tube for reaction. The electric heating system could control the furnace temperature through a temperature control program. The inner diameter of the corundum tube was 30 mm and the length of constant-temperature zone was 290 mm.

The amount of NO produced by combustion was determined by the testo 335 gas analyzer with a range of 0–2000 μL/L and an accuracy of ±10% for 0~99 μL/L and ±5% for 100~1999 μL/L. Each case was repeated three times and the root-mean-square error was used to represent the test error. One end of the gas analyzer was connected to the exhaust gas and the other end was connected to a computer, and the exhaust gas composition data was monitored by testo easyEmission software.

### 2.2. Experiment Cases

The working conditions of this experiment are shown in Table 1. In this experiment, no $N_2$ was involved and Ar was used as the background gas to dilute the mixed gas, so the NO produced by combustion was fuel $NO_x$ rather than the fast type or thermal type. Therefore, there was no setting of pure methane or pure carbon monoxide combustion conditions in the experiment. When the excess air coefficient was lower than 0.9, the output of NO was very low, so the experimental situation when the excess air coefficient was lower than 0.9 was also excluded. Influencing factors such as ammonia ratio in the fuel gas, excess air coefficient, temperature, residence time, and fuel concentration were changed in the experiment.

**Table 1.** Operation settings.

| Influencing Factors | | Setting |
|---|---|---|
| Fuel volume fraction | $NH_3$/% | 0.83–2.78 |
| | $CH_4$/% | 0–1.94 |
| | $NH_3$/% | 0.69–2.78 |
| | CO/% | 0–2.08 |
| Ammonia ratio in fuel gas/% | | 25–100 |
| Excess air coefficient | | 0.9–1.1 |
| Temperature/°C | | 900–1500 |
| Residence time/ms | | 565–3146 |

## 3. Results and Discussion

### 3.1. NO Formation in Pure Ammonia Combustion

3.1.1. Effect of Residence Time *t* on NO Production

Combustion experiments were conducted for a certain proportion of $NH_3/O_2/Ar$ at different residence times and NO content in the flue gas was measured. It was predicted that the longer the residence time of the mixed gas in the high-temperature furnace, the more fully the $NH_3$, $O_2$, and Ar would be mixed, and the higher the degree of ammonia combustion. The experimental results are shown in Figure 2. The excess air coefficients were set as 0.9, 1.0, and 1.1; the combustion temperature was set as 1500 °C; and the residence time varied from 565 ms to 2258 ms. As shown in Figure 2, when $\alpha$ was not less than 1, the amount of NO produced by ammonia combustion decreased with the increase in residence time, which was basically consistent with the prediction. The longer the residence time, the fuller the combustion, and the less NO produced. With the increase in residence time, the influence of residence time on NO generation also gradually decreased. However, when $\alpha$ was less than 1, it can be seen that the NO amount increased when the residence time was 1129ms. This may have been because the ratio of $NH_3$ to $O_2$ increases in the case of rich combustion, resulting in an increase in incomplete combustion, and hence the NO amount also increased.

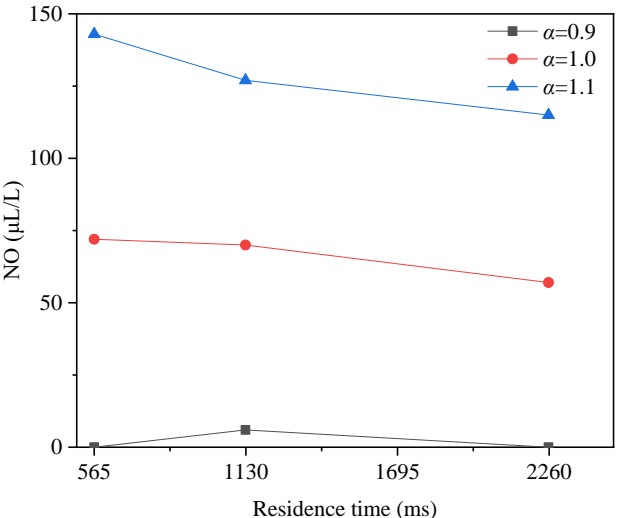

**Figure 2.** NO produced by $NH_3$ combustion at different residence times at 1500 °C.

### 3.1.2. Effect of Ammonia Concentration on NO Production

In this section, the ammonia concentration was doubled by doubling the $NH_3$ flow rate in the $NH_3/O_2/Ar$ mixture and the $O_2$ flow rate corresponding to the excess air coefficient at the same time. The excess air coefficients were 0.9, 1.0, and 1.1, and the combustion temperature was 1500 °C. The experimental results are shown in Figure 3. Under several of the temperature conditions set in this experiment, NO production also increased when $\alpha$ was not less than 1 and the ammonia concentration doubled, and the influence of concentration-doubling increased with the increase in excess air coefficient. When $\alpha$ was less than 1, the NO production was always low. When $\alpha$ was smaller, the competition between $H + O_2 + M = HO_2 + M$ and $H + O_2 = O + OH$ reactions was more intense, and the tendency to prevent the generation of O/H radicals increased, which inhibited the conversion of $NH_3$ to $NH_2$ and ultimately inhibit the reduction of NO [32]. Hence NO production was invariably low when $\alpha$ is less than 1.

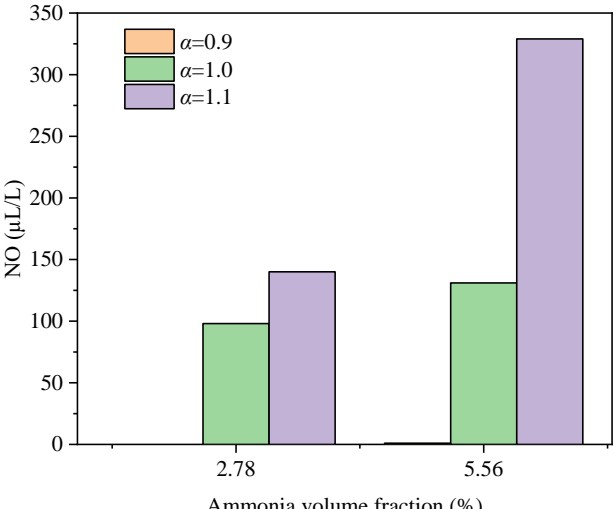

**Figure 3.** NO produced at different fuel gas concentrations at 1500 °C.

### 3.2. NO Formation Characteristics of $NH_3/CH_4$ Combustion

#### 3.2.1. Effect of Proportion of $NH_3$ in Fuel Gas on NO Production

In this section, we describe our investigation of the effect of the proportion of ammonia in $NH_3/CH_4$ on the emission of NO from combustion at different combustion temperatures. Figure 4 shows the curve of NO production with different proportions of ammonia in fuel

gas at different combustion temperatures. The proportion of ammonia gas varied from 30% to 100%; the combustion temperatures were 900 °C, 1100 °C, 1300 °C, and 1500 °C; and the excess air coefficients were 0.9, 1.0, and 1.1. As shown in Figure 4, when the excess air coefficient was 0.9, the amount of NO production was very low, and it was almost zero above 1100 °C.

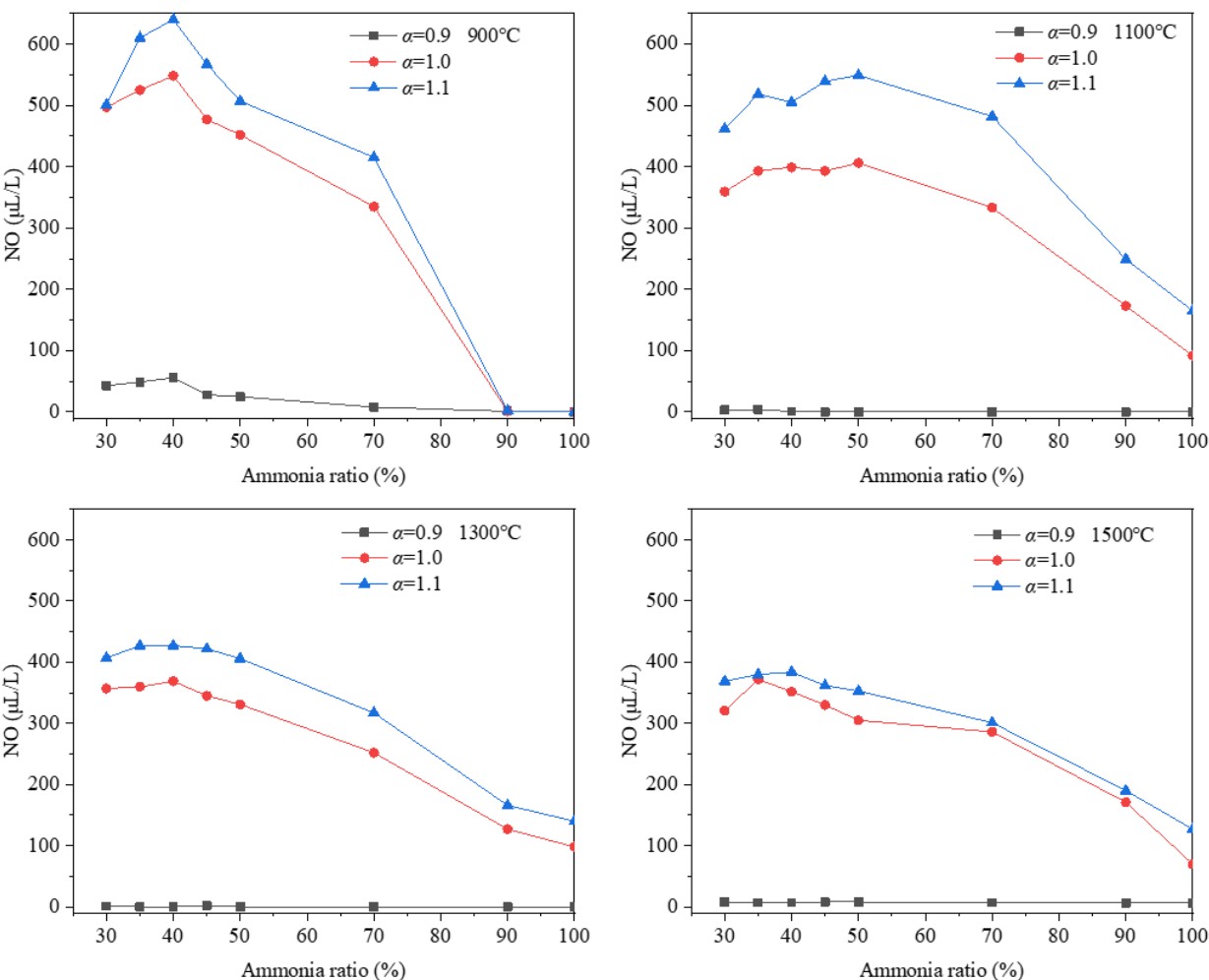

**Figure 4.** NO production of $NH_3/CH_4$ mixed firing under different ammonia ratios.

With the temperature and excess air coefficient set in the experiment, the amount of NO produced by $NH_3/CH_4$ blended gas combustion first increased and then decreased with the increase in the proportion of ammonia in the fuel gas, and the emission of NO reached the maximum when the proportion of ammonia was about 40%. Xiao et al. [33] and Kurata [34] found similar results. In $NH_3$-$CH_4$-air combustion, NO emissions increased sharply with the increase in $NH_3$ fuel ratio. However, from a certain point near the $NH_3$ fuel ratio of 0.65, NO decreased as the $NH_3$ fuel ratio further increased. Fenimore [35] showed that high levels of fuel nitrogen led to low fractions of NO.

When the excess air coefficient $\alpha \geq 1$, the main production path of amino $NH_2$ is [36]

$$NH_3 + OH = NH_2 + H_2O \tag{1}$$

$$NH_3 + O = NH_2 + OH \tag{2}$$

The reaction product, $NH_2$ free radical, reacts with O atom to produce an important intermediate HNO, which is the main way that HNO is formed under dilute combustion conditions, and the HNO group is subsequently converted to produce NO. When the

proportion of ammonia is less than 40%, the main rate-limiting reaction of ammonia oxidation path in methane-ammonia flame is [23]

$$NO + NH_2 = NNH + OH \tag{3}$$

$$NH_2 + O = HNO + H \tag{4}$$

$$HNO + H = NO + H_2 \tag{5}$$

In the ammonia oxidation process, Reactions (3)–(5) exert significant influence on the concentration of H and OH radicals in the methane–ammonia flame. When the methane concentration decreases, the ammonia concentration increases, resulting in higher concentrations of H and OH radicals. This promotes the reverse progression of Reactions (3) and (4), leading to an increase in NO production.

When the proportion of ammonia in the fuel gas exceeds 40%, the reducibility of ammonia to NO becomes prominent. Thus, an increase in ammonia concentration promotes the formation of the $NH_x$ group and facilitates the reduction of NO to $N_2$. The primary reactions taking place during this process are as follows:

$$NH_2 + NO = NNH + OH \tag{6}$$

$$NNH + NO = N_2 + HNO \tag{7}$$

$$NH_2 + NO = N_2 + H_2O \tag{8}$$

$$NH + NO = N_2O + H \tag{9}$$

$$N_2O + H = N_2 + OH \tag{10}$$

Reactions (6) and (8) are the key to the mechanism of thermal NO removal (selective non-catalytic reduction of NO) in ammonia flame chemistry [36], and these NO reburning reactions are not associated with flames containing only trace amounts of ammonia. With the increase in $NH_3$ proportion and concentration, the positive occurrence of Reaction (6) and (8) is promoted, resulting in the decrease in NO production.

The main oxidation pathway of methane is [37]

$$CH_4 \rightarrow CH_3 \rightarrow CH_3O \rightarrow CH_2O \rightarrow HCO \rightarrow CO \rightarrow CO_2 \tag{11}$$

$CH_i$, which is produced in large quantities in the first step, is also the main intermediate for the formation of rapid NO. As the concentration of ammonia increases, the concentration of methane decreases and the $CH_i$ group decreases, which further reduces the NO produced.

When the excess air coefficient $\alpha < 1$, NO emissions are at a low level, which will be discussed in Section 3.2.2.

### 3.2.2. Effect of Excess Air Coefficient $\alpha$ on NO Production

This section focuses on the effect of $NH_3/CH_4$ combustion on NO emissions under different excess air coefficients. As shown in Figure 4, the emission of NO from combustion changed little when the excess air coefficient was greater than 1, but decreased significantly when the excess air coefficient was less than 1.

Each $NH_i$ radical has two subsequent reactions: oxidation to NO or reaction with NO to $N_2$. The determination of the $NO/N_2$ product distribution is primarily influenced by the chemical identity of the $NH_i$ free radical, which is dependent on the equivalence ratio [36]. As the excess air coefficient decreases, the abundance of hydrogen atoms increases, resulting in a transition of the critical $NH_i$ from $NH_2$ to NH to N. This explains why the smaller the excess air coefficient, the less NO was produced by combustion.

When the excess air coefficient $\alpha \geq 1$, the effect of the excess air coefficient on NO production was relatively small. When the excess air coefficient was 1.1, the emission of NO increased relative to the chemical equivalent ratio, but the amplitude was small. This may be the reason that, with the increase in the excess air coefficient, the transformation of

the critical amino group to NO was intensified, the concentration of hydrogen atom was smaller, and the increase in NO was smaller. However, when the excess air coefficient was 0.9, the conversion and production of NO decreased rapidly compared with the chemical equivalent ratio.

When the excess air coefficient $\alpha < 1$, on the one hand, the methane in the fuel gas may undergo incomplete combustion to produce CO, and the reduction reaction with NO occurs; on the other hand, the concentration of O/H free radicals decreases during the combustion of ammonia under the condition of rich combustion, and the proportion of H in O/H free radicals increases, resulting in a large number of H free radicals. This promotes $NH_x$ (x = 1, 2, 3) free radicals to bind to H and oxidize. The main reactions that occur during this process are as follows:

$$NH_3 + H = NH_2 + H_2 \tag{12}$$

$$NH_2 + H = NH + H_2 \tag{13}$$

$$NH + H = N + H_2 \tag{14}$$

The $NH_x$ (x = 1, 2, 3) bonding reaction is dominant in rich combustion. Under this reaction condition, ammonia is mainly converted into $N_2$ through NNH without producing NO. Therefore, under the conditions set in this experiment, the emission of NO generated by combustion was very low when the excess air coefficient $\alpha < 1$.

### 3.2.3. Effect of Temperature $T$ on NO Production

This section details the effects of different combustion temperatures on the amount of NO produced by $NH_3/CH_4$ combustion under different excess air coefficients. The experimental results are shown in Figure 5. As was determined in Section 3.2.1 that when ammonia accounted for about 40% of mixed fired gas, NO emissions reached the maximum value. Therefore, we used 40% ammonia in mixed fuel gas for the experiment and the combustion temperatures were 900 °C, 1100 °C, 1300 °C, and 1500 °C. Excess air coefficients were 0.9, 1.0, and 1.1.

As shown in Figure 5, under the three excess air coefficients, NO emission decreased with the increase in temperature, and the decline rate of NO emission slowed down with the gradual increase in temperature. Within a certain temperature range, $NH_3$ can enhance the reduction of $NO_x$ through the thermal denitration process, and Reactions (6) and (8) occur.

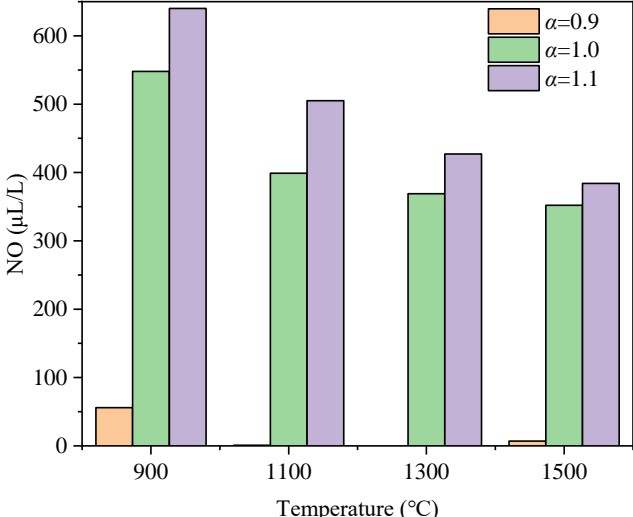

**Figure 5.** The content of NO in combustion flue gas at different temperatures.

At the end of Reaction (8), NO is reduced to nitrogen, and the increase in temperature reinforces this step, leading to a decrease in NO production with increasing temperature.

With the increase in temperature, the branching ratio gradually increases, resulting in the increase in NNH production, then resulting in the following processes:

$$NNH = N_2 + H \tag{15}$$

$$H + O_2 = O + OH \tag{16}$$

$$O + H_2O = OH + OH \tag{17}$$

Since NNH is short-lived, the amino group will react with the O/H radical to form NO. Reactions of (6) and (8) are gradually balanced, so that the decline rate of NO production gradually slows down.

### 3.2.4. Effect of Residence Time *t* on NO Production

The research content of this section is about the influence of residence time in the high-temperature furnace on NO emission when the mixed fuel gas with 50% ammonia gas is burned at different temperatures and different excess air coefficients. The residence time of mixed gas in high-temperature furnace mainly affects the mixing degree of four gases and the combustion degree of fuel gas in the furnace. The experimental results are shown in Figure 6. The combustion temperature was set to 1500 °C; the excess air coefficients were 0.9, 1.0, and 1.1; and the residence time varied from 565 ms to 2258 ms.

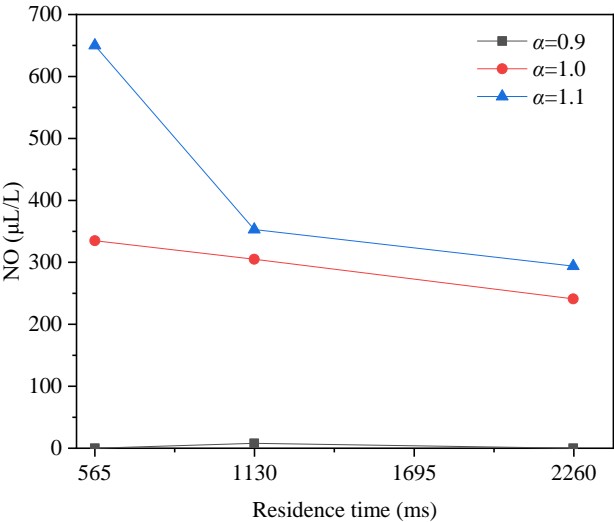

**Figure 6.** NO produced by $NH_3/CH_4$ combustion at different residence times at 1500 °C.

As shown in Figure 6, the amount of NO produced by combustion gradually decreased with the increase in residence time, which was in line with our general understanding: the longer the residence time, the more fully the gas is mixed in the high-temperature furnace, the deeper the combustion degree of the fuel gas in the furnace, the fewer the O/H free radicals generated during the combustion process, thus reducing the oxidation of amino $NH_2$ and reducing the emission of NO. It is worth noting that with the increase in residence time, the decrease in NO content produced by combustion also gradually decreased. Due to the key free radicals produced in the ammonia reaction not being sensitive to the residence time, the O/H free radicals in the combustion furnace reached a relative equilibrium state with the extension of the residence time after full reaction, and the content of NO tended to be stable.

It can be seen from Section 3.2.1 that the combustion of doped gas produced more NO than the combustion of pure ammonia, which was due to the large increase in HNO, $NH_i$, $N_2O$, and other free radicals generated by the reaction under the mixed combustion condition, and the increased selectivity of $NH_2$ free radical conversion to NO. This makes the amount of NO produced by $NH_3/CH_4$ mixed combustion gas in any case more than that of pure ammonia combustion.

### 3.2.5. Effect of Gas Concentration on NO Production

We investigated the influence of $NH_3/CH_4$ doped gas concentration changes on the amount of NO production under different excess air coefficients and combustion temperatures. The experimental results are shown in Figure 7. The proportion of ammonia in the mixed combustion gas used was 50%; the excess air coefficients were set at 0.9, 1.0, and 1.1; and the combustion temperature was set at 1500 °C. When $\alpha \geq 1$, the amount of NO produced by combustion increased when the fuel gas concentration was doubled. When the excess air coefficient was 1.0, the increase rate was small, and when the excess air coefficient reached 1.1, the amount of NO emissions increased significantly. After the concentration of gas was doubled, the free radicals such as HNO and OH produced by the reaction increased, so that the amount of NO produced also increased. However, when $\alpha < 1$ and the fuel concentration increased, NO production changed from a minimal amount to zero. This may be the reason that, when the excess air coefficient $\alpha < 1$, the methane in the fuel gas may undergo incomplete combustion to produce CO, and the reduction reaction with NO occurs. On the other hand, the concentration of O/H free radicals decreases during the combustion of ammonia under the condition of rich combustion, and the proportion of H in O/H free radicals increases, resulting in a large number of H free radicals. This promotes the binding of $NH_x$ ($x = 1, 2, 3$) free radicals to H and its oxidation (Reactions 12–14). NO production is too low, hence the change is not obvious and difficult to observe.

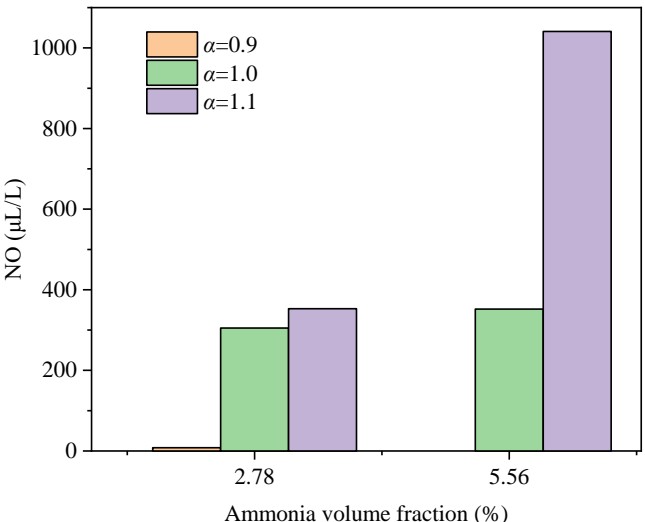

**Figure 7.** NO production of $NH_3/CH_4$ co-firing under different fuel gas concentrations at 1500 °C.

### 3.3. NO Formation Characteristics of $NH_3/CO$ Combustion

### 3.3.1. Effect of Proportion of $NH_3$ in Fuel Gas on NO Production

In this section, we describe the effects of different excess air coefficients and the proportion of ammonia in $NH_3/CO$ on the amount of NO produced by combustion at different temperatures. The change curve of NO content in combustion flue gas with the proportion of $NH_3$ in fuel gas is shown in Figure 8. The excess air coefficients were set to 0.9, 1.0, and 1.1; the combustion temperatures were set to 900 °C, 1100 °C, 1300 °C, and 1500 °C; and the proportion of ammonia varied from 25% to 100%.

As shown in Figure 8, when the excess air coefficient $\alpha \geq 1$, NO generation increased first and then decreased with the increase in the proportion of ammonia in fuel gas. When the proportion of ammonia was about 50%, NO generation reached the maximum. When the excess air coefficient $\alpha < 1$, the production of NO gradually decreased with the increase in ammonia proportion.

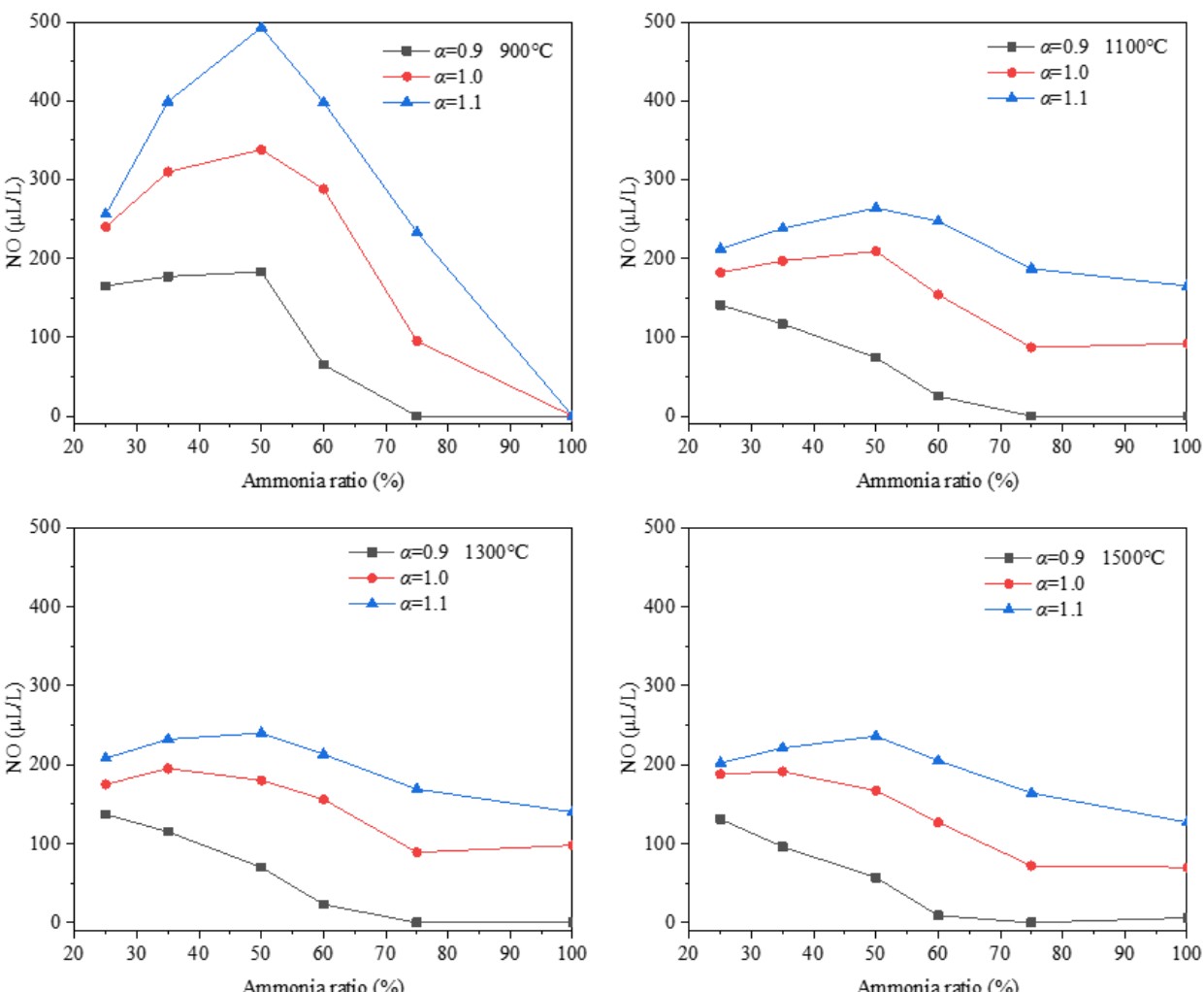

**Figure 8.** NO production in $NH_3/CO$ combustion under different ammonia ratios.

When $\alpha \geq 1$ and ammonia ratio was less than 50%, and the CO reaction played a very important role. At very high CO concentrations, carbon monoxide is oxidized almost entirely by reacting with OH radicals [38]

$$CO + OH = CO_2 + H \tag{18}$$

The hydrogen atoms formed in this reaction can be used to oxidize ammonia,

$$NH_3 + H = NH_2 + H_2 \tag{19}$$

Therefore, under these conditions, CO and $NH_3$ do not compete for the same free radicals. The presence of high levels of CO promotes, rather than inhibits, the conversion of $NH_3$ and NO to $N_2$.

In addition, a high concentration of CO promotes the formation of HNCO through the $NH_2$+CO reaction. However, due to the decrease in the concentration of O and H radicals, the reaction of $NH_2$ + O to generate HNO is inhibited, and the production of NO is further reduced [39]. Skreiberg et al. [38] also noted that adding CO to a lean $NH_3/NO/O_2/N_2$ flame at 1000 °C increased the O/H radical pool, thereby increasing the amount of NO produced via the HNO pathway.

### 3.3.2. Effect of Excess Air Coefficient $\alpha$ on NO Production

We investigated the influence of the excess air coefficient on NO generation characteristics during $NH_3/CO$ combustion at different combustion temperatures. The excess air coefficients were set as 0.9, 1.0, and 1.1, and the furnace temperature varied from 900 °C to 1500 °C. As shown in Figure 8, at the combustion temperature set in the experiment, the NO content in the flue gas generated by combustion gradually increased with the increase in excess air coefficient $\alpha$.

### 3.3.3. Effect of Temperature $T$ on NO Production

The effect of combustion temperature on the amount of NO produced by combustion under different excess air coefficients was investigated. The combustion temperature ranged from 900 °C to 1500 °C, and the excess air coefficients were 0.9, 1.0, and 1.1. It can be seen from the results discussed in Section 3.3.1 that under certain conditions of excess air coefficient and combustion temperature, when ammonia accounted for about 50% of the fuel gas, the amount of NO produced reached its peak. Therefore, to conduct the experiment, the proportion of ammonia gas in the blended fuel gas was 50%. The influence curve of combustion temperature on NO generation characteristics is shown in Figure 9. With the increase in combustion temperature, the amount of NO generated in flue gas gradually decreased, and the higher the temperature, the lower the influence on NO generation.

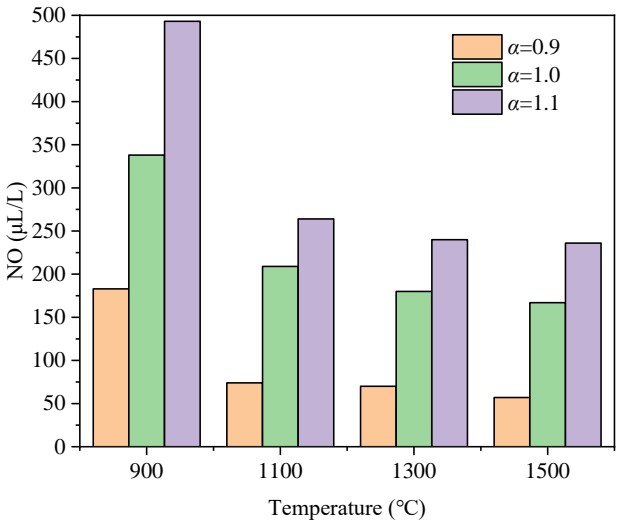

**Figure 9.** The content of NO in combustion flue gas varies with temperature.

### 3.3.4. Effect of Residence Time $t$ on NO Production

To investigate the effect on residence time on NO production, the combustion temperature was 1500 °C; the excess air coefficients were 0.9, 1.0, and 1.1; and the residence times varied from 565 ms to 2258 ms. The experimental result is shown in Figure 10. With the gradual increase in residence time, the amount of NO produced by combustion gradually decreased, and with the decrease in excess air coefficient or a gradual increase in residence time, the influence of NO amount on residence time also gradually reduced.

### 3.3.5. Effect of Gas Concentration on NO Production

The influence of $NH_3/CO$ doped gas concentration changes on the amount of NO under different excess air coefficients and combustion temperatures was investigated. The experimental results are shown in Figure 11. The proportion of ammonia in the mixed combustion gas was 50%; the excess air coefficients were set at 0.9, 1.0, and 1.1; and the combustion temperature was set at 1500 °C. As shown in Figure 11, when the excess air coefficient $\alpha \geq 1$, the amount of NO produced by combustion also increased with the

increase in the fuel concentration, and the influence of the concentration was also enhanced by an increase in the excess air coefficient. This is because the OH group and HNO group produced by the doubling of the fuel concentration in the lean combustion state increased, and they were further converted into NO. However, when the excess air coefficient $\alpha < 1$ increased the fuel concentration, the production of NO decreased instead, because the incomplete combustion of CO and $NH_3$ will participate in the reduction reaction of NO in the case of rich combustion, so the production of NO will decline.

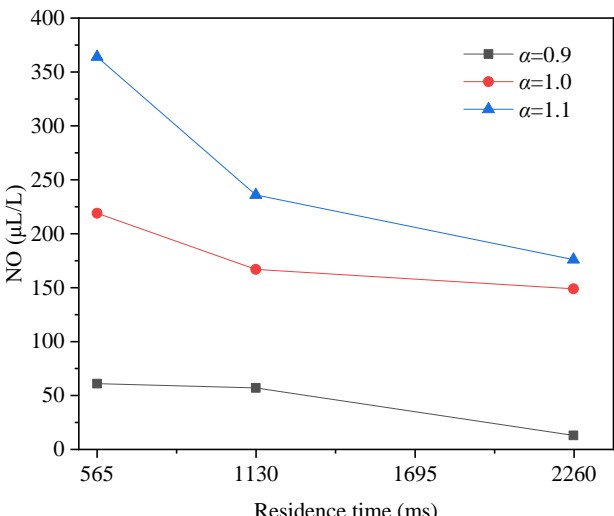

**Figure 10.** NO produced by $NH_3/CO$ combustion at different residence times at 1500 °C.

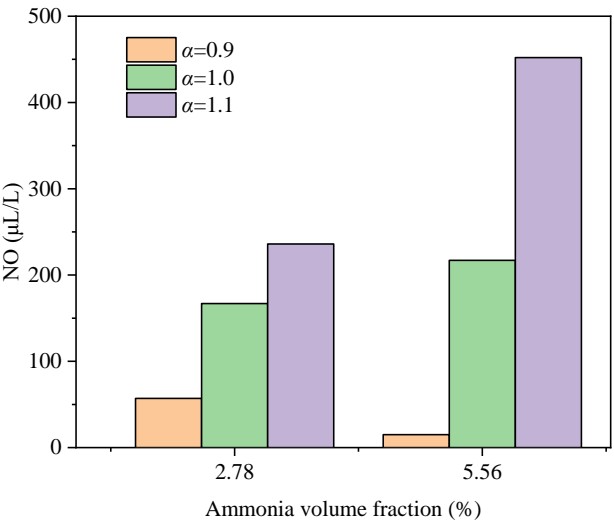

**Figure 11.** NO production of $NH_3/CO$ co-firing under different fuel gas concentrations at 1500 °C.

## 4. Conclusions

In this study, the effects of ammonia ratio, excess air coefficient, temperature, residence time, and gas concentration on NO generation during the combustion of $NH_3/CH_4$ and $NH_3/CO$ were investigated. The main conclusions are as follows:

1.  When the excess air coefficient $\alpha \geq 1$, with the increase in the proportion of ammonia in the fuel gas, the production of NO increased first and then decreased. When the proportion of ammonia in $NH_3/CH_4$ was 40% or 50% in $NH_3/CO$, the production of NO reached the maximum value. However, when the excess air coefficient $\alpha < 1$, the NO production gradually decreased with the increase in ammonia proportion. The NO emissions of the blended fuel were higher than that of pure ammonia.

2.  When the excess air coefficient $\alpha$ was in the range of 0.9–1.1, the emission of NO from combustion increased with the increase in the excess air coefficient, but the excess air coefficient had little effect on the NO production when $\alpha \geq 1$, and the NO production when $\alpha < 1$ was at a very low level when $NH_3/CH_4$ was mixed.
3.  In the range of 900–1500 °C, the amount of NO produced by mixed gas combustion gradually decreased with the increase in temperature, and the degree of influence of temperature on the amount of NO production gradually weakened with the increase in temperature.
4.  The amount of NO produced by combustion gradually decreased with the increase in residence time, and, according to the change trend of the amount of NO, when the residence time was long enough, the NO content in the flue gas was in a stable state. The increase in residence time was associated with the decrease in NO production, and the effect of residence time on NO production was gradually weakened with the increase in residence time.
5.  The concentration of fuel gas in the mixture had a large influence on the generation of fuel $NO_x$. When the excess air coefficient $\alpha \geq 1$, the amount of NO produced by combustion also increased with the increase in the fuel concentration, and the influence of the concentration was also enhanced with the increase in the excess air coefficient. However, when the excess air coefficient $\alpha < 1$ increased the fuel concentration, NO production decreased.

Based on the analysis of experimental results and comparison with other literature, we believe that these results can be extrapolated to different combustion scenarios or fuels. These findings have potential implications for optimizing combustion processes in terms of NO emission control. With suitable conditions, the fuel ammonia combustion can be controlled and produce lower NO emissions. For example, the excess air coefficient should not be too high, and the residence time should not be too short. The $NO_x$ formation characteristics of ammonia combustion mixed with methane or carbon monoxide was studied in this paper. This will help guide the clean and efficient utilization of ammonia, which is a new renewable energy source, and which can turn industrial by-products into wealth. Based on the insights gained from this paper, the key area of future research should be how to improve the combustion efficiency of ammonia and reduce NO emissions by mixing industrial byproducts such as blast furnace gas or low calorific value gas while reducing the cost, so as to achieve complementary advantages.

**Author Contributions:** Conceptualization, Y.D. (Yongbo Du) and C.W.; formal analysis, S.Z., C.W. and Y.W.; funding acquisition, Y.D. (Yongbo Du); investigation, S.Z. and Y.W.; project administration, Y.D. (Yongbo Du); supervision, Q.L., Y.D. (Yaodong Da) and D.C.; writing—original draft, Y.D. (Yongbo Du) and S.Z.; writing—review and editing, Y.D. (Yongbo Du), Q.L., Y.D. (Yaodong Da) and D.C. All authors have read and agreed to the published version of the manuscript.

**Funding:** The authors acknowledge financial support from the National Key R&D Program of China (2021YFF0600601) and the Shaanxi Provincial Technology Innovation Guidance Special Project (2022QFY06-02), China.

**Institutional Review Board Statement:** Not applicable.

**Informed Consent Statement:** Not applicable.

**Data Availability Statement:** The data presented in this study are available on request from the corresponding author. The data are not publicly available due to [privacy].

**Acknowledgments:** The authors would like to thank Zhang for his help and his conscientious work.

**Conflicts of Interest:** The authors declare no conflict of interest.

## Nomenclature

*Symbols*
| | |
|---|---|
| $T$ | Temperature |
| $t$ | Residence time |

*Greek symbols*
| | |
|---|---|
| $\alpha$ | Excess air coefficient |

*Acronyms*
| | |
|---|---|
| $NO_x$ | Nitrogen oxide |
| $SO_x$ | Sulfur oxide |

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
