# Peer review of "Study on Mechanisms of NOx Formation and Inhibition during the Combustion of NH3/CH4 and NH3/CO Mixtures"

_applsci, doi:10.3390/app132111847_

Round 1

Reviewer 1 Report

Comments and Suggestions for Authors

The proposed study shows the results of an experimental analysis carried out to assess the mechanisms of NOx formation and inhibition during the combustion of NH3/CH4 and NH3/CO mixtures. The paper is generally well written and the methodology is well described. Anyway, there are some parts of the Methodology and Results sections that need to be improved. In the following some specific comments are provided.

1.       Section 2.1. The sentence “Use flow meter and D08-3E flow display to control air flow with ±0.5% accuracy” needs to be rewritten.

2.       Section 2.1. “results are averaged over many measurements”, please specify the number of measurements for each test case.

3.       Section 3.2.1. The description of the results of references [35], [36] and [37] should be shortened in this section.

4.       Sections 3.3.1-3.3.3. No data/graphs are reported referring to the effect of the discussed parameter on NO formation. They need to be added.

Reviewer 2 Report

Comments and Suggestions for Authors

1. What is the impact of ammonia proportion in the fuel gas on NO production when the excess air coefficient is ≥1, and how does it vary with different ammonia proportions?

2. Why does NO production increase first and then decrease with the increase in ammonia proportion in NH3/CH4 when excess air coefficient α≥1?

3. How does the excess air coefficient α affect NO production in the context of ammonia and methane mixture, and why does it have little effect when α≥1?

4. What are the reasons for the low level of NO production when α<1 and NH3/CH4 are mixed?

5. How does temperature influence NO production in mixed gas combustion, and what is the relationship between temperature and NO emissions?

6. Why does the degree of temperature's influence on NO production weaken with increasing temperature?

7. How does residence time impact NO production during combustion, and what is the trend observed as residence time increases?

8. What are the factors contributing to the stabilization of NO content in flue gas with long residence times?

9. How does the effect of residence time on NO production change as residence time increases?

10. How does the concentration of fuel gas in the mixture affect the generation of fuel NOx, especially when the excess air coefficient is ≥1?

11. What is the relationship between fuel concentration and NO production when α≥1?

12. Why does NO production increase with the fuel concentration, and how does the excess air coefficient influence this relationship?

13. What happens to NO production when the excess air coefficient α<1 and the fuel concentration is increased?

14. Are there specific mechanisms or chemical reactions responsible for the observed trends in NO production in this context?

15. How do these findings relate to practical applications and environmental concerns regarding NO emissions?

16. Can these results be extrapolated to different combustion scenarios or fuels, or are they specific to the conditions described in the context?

17. What are the potential implications of these findings for optimizing combustion processes in terms of NO emission control?

18. Are there any technological or engineering solutions that can be derived from this research to mitigate NO emissions in practical applications?

19. How do the findings in this context contribute to our understanding of NOx emissions in the broader field of combustion and environmental science?

20. What are the key areas for future research or experimentation based on the insights gained from this study?
21. Kindly some of recent literatures about NOx capture and experimental work as mentioned below:

Ramesh, T., A. P. Sathiyagnanam, Melvin Victor De Poures, and P. Murugan. "A comprehensive study on the effect of dimethyl carbonate oxygenate and EGR on emission reduction, combustion analysis, and performance enhancement of a CRDI diesel engine using a blend of diesel and Prosopis juliflora biodiesel." International Journal of Chemical Engineering 2022 (2022).

Ramesh, T., A. P. Sathiyagnanam, Melvin Victor De Poures, and P. Murugan. "Combined effect of compression ratio and fuel injection pressure on CI engine equipped with CRDi system using Prosopis juliflora methyl ester/diesel blends." International Journal of Chemical Engineering 2022 (2022): 1-12.

Comments on the Quality of English Language

1. What is the impact of ammonia proportion in the fuel gas on NO production when the excess air coefficient is ≥1, and how does it vary with different ammonia proportions?

2. Why does NO production increase first and then decrease with the increase in ammonia proportion in NH3/CH4 when excess air coefficient α≥1?

3. How does the excess air coefficient α affect NO production in the context of ammonia and methane mixture, and why does it have little effect when α≥1?

4. What are the reasons for the low level of NO production when α<1 and NH3/CH4 are mixed?

5. How does temperature influence NO production in mixed gas combustion, and what is the relationship between temperature and NO emissions?

6. Why does the degree of temperature's influence on NO production weaken with increasing temperature?

7. How does residence time impact NO production during combustion, and what is the trend observed as residence time increases?

8. What are the factors contributing to the stabilization of NO content in flue gas with long residence times?

9. How does the effect of residence time on NO production change as residence time increases?

10. How does the concentration of fuel gas in the mixture affect the generation of fuel NOx, especially when the excess air coefficient is ≥1?

11. What is the relationship between fuel concentration and NO production when α≥1?

12. Why does NO production increase with the fuel concentration, and how does the excess air coefficient influence this relationship?

13. What happens to NO production when the excess air coefficient α<1 and the fuel concentration is increased?

14. Are there specific mechanisms or chemical reactions responsible for the observed trends in NO production in this context?

15. How do these findings relate to practical applications and environmental concerns regarding NO emissions?

16. Can these results be extrapolated to different combustion scenarios or fuels, or are they specific to the conditions described in the context?

17. What are the potential implications of these findings for optimizing combustion processes in terms of NO emission control?

18. Are there any technological or engineering solutions that can be derived from this research to mitigate NO emissions in practical applications?

19. How do the findings in this context contribute to our understanding of NOx emissions in the broader field of combustion and environmental science?

20. What are the key areas for future research or experimentation based on the insights gained from this study?
21. Kindly some of recent literatures about NOx capture and experimental work as mentioned below:

Ramesh, T., A. P. Sathiyagnanam, Melvin Victor De Poures, and P. Murugan. "A comprehensive study on the effect of dimethyl carbonate oxygenate and EGR on emission reduction, combustion analysis, and performance enhancement of a CRDI diesel engine using a blend of diesel and Prosopis juliflora biodiesel." International Journal of Chemical Engineering 2022 (2022).

Ramesh, T., A. P. Sathiyagnanam, Melvin Victor De Poures, and P. Murugan. "Combined effect of compression ratio and fuel injection pressure on CI engine equipped with CRDi system using Prosopis juliflora methyl ester/diesel blends." International Journal of Chemical Engineering 2022 (2022): 1-12.

Reviewer 3 Report

Comments and Suggestions for Authors

The manuscript scope is dealing with investigation the mechanisms of NOx formation and inhibition during the combustion of NH3/CH4 and NH3/CO mixtures. However, there are some comment may improve the quality of the manuscript such as the following:

1.       Bulky references in introduction section should be avoided

2.      The Nomenclature and symbols section should be added

3.       The description of the uncertainty analysis should be added to determine the total error of the experimental data  

4.       Figure 1 should including a direct photo for the test layout and the measuring system

5.       All the obtained result figure cooler quality should be enhanced

6.       The conclusion section should be revised

Round 2

Reviewer 1 Report

Comments and Suggestions for Authors

The manuscript was revised according to the remarks provided and can be accepted in the present form.